# Combinatorial biosynthesis for the engineering of novel fungal natural products
**Elizabeth Skellam** [1,2,3] ✉**, Sanjeevan Rajendran** [1,2] **& Lei Li**[1,2]

Natural products are small molecules synthesized by fungi, bacteria and plants, which historically have had a profound effect on human health and quality of life. These natural products have evolved over millions of years resulting in specific biological functions that may be of interest for pharmaceutical, agricultural, or nutraceutical use. Often natural products need to be structurally modified to make them suitable for specific applications. Combinatorial biosynthesis is a method to alter the composition of enzymes needed to synthesize a specific natural product resulting in structurally diversified molecules. In this review we discuss different approaches for combinatorial biosynthesis of natural products via engineering fungal enzymes and biosynthetic pathways. We highlight the biosynthetic knowledge gained from these studies and provide examples of new-to-nature bioactive molecules, including molecules synthesized using combinations of fungal and non-fungal enzymes.

Fungi have long been recognized as prolific producers of bioactive secondary metabolites, commonly referred to as natural products, including antibiotics[1,2], immunosuppressants[3,4], and anticancer agents[5,6], amongst others (Fig. 1). Natural products are classified according to their biosynthetic origin exemplified by polyketides *e.g.* **1**–**3**, non-ribosomal peptides *e.g.* **4**–**6**, hybrid molecules *e.g.* **7** and **8**, and terpenes *e.g.* **9** and **10**, (Fig. 1). Hybrid natural products additionally include polyketide-terpene hybrids (meroterpenoids), peptide-terpene hybrids, and glycosylated molecules, significantly expanding the structural diversity of these compounds. Fungal natural products are produced *via* highly programmed pathways and originate from simple building blocks[7] such as acyl-CoAs, proteinogenic- and non-proteinogenic amino acids, isopentenyl-pyrophosphate (IPP)/dimethylallylpyrophosphate (DMAPP), and sugars. From a biosynthetic point of view, the diversity and complexity of natural products is generated in a two-step process: (i) the formation of the core hydrocarbon scaffold and (ii) the modification of this scaffold by tailoring enzymes.

Fungi and their complex biosynthetic machinery have evolved to synthesize a wide variety of bioactive natural products, making them a valuable resource for drug discovery[8,9]. However, the potential of fungi is sometimes limited by certain compounds being produced only in low amounts by specific strains that are difficult to cultivate or engineer. Benefiting from high throughput genome sequencing, accumulated understanding of biosynthetic pathways, advancements in synthetic biology tools, and development of heterologous hosts, combinatorial biosynthesis has

become a robust strategy for accessing natural products with novel structures and biological activities. The concept of combinatorial biosynthesis[10] involves expanding the biosynthetic inventory of a fungal producer by introducing non-native enzymes into specific biosynthetic pathways, ultimately manipulating the natural product output. The novel compounds generated hold great promise in the quest for new drugs and therapeutic agents, and the approach provides valuable tools for addressing health challenges, the future energy crisis, and advancing our understanding of natural product biosynthesis.

This review covers the various approaches used for combinatorial biosynthesis of fungal natural products highlighting studies that have generated novel molecules, created molecules with new biological activities, and led to enhanced understanding of enzyme programming or mechanistic features.

## Combinatorial biosynthesis of megasynth(et)ases

Megasynth(et)ases are the large, multifunctional enzymes that synthesize the essential carbon framework of a natural product. There are two main approaches to combine biosynthetic features from different megasynth(et)ases: (i) domain swaps and (ii) module swaps[11]. In the domain swap approach the megasynth(et)ase is deconstructed into individual domains and recombined with analogous domains, generating a chimeric enzyme. Some types of fungal enzymes contain domains organized into modules, where the entire module can be swapped with analogous modules to create new enzyme combinations.

[1]Department of Chemistry, University of North Texas, 1155 Union Circle, Denton, TX 76203, USA. [2]BioDiscovery Institute, University of North Texas, 1155 Union Circle, Denton, TX 76203, USA. [3]Department of Biological Sciences, University of North Texas, 1155 Union Circle, Denton, TX 76203, USA. ✉e-mail: elizabeth.skellam@unt.edu

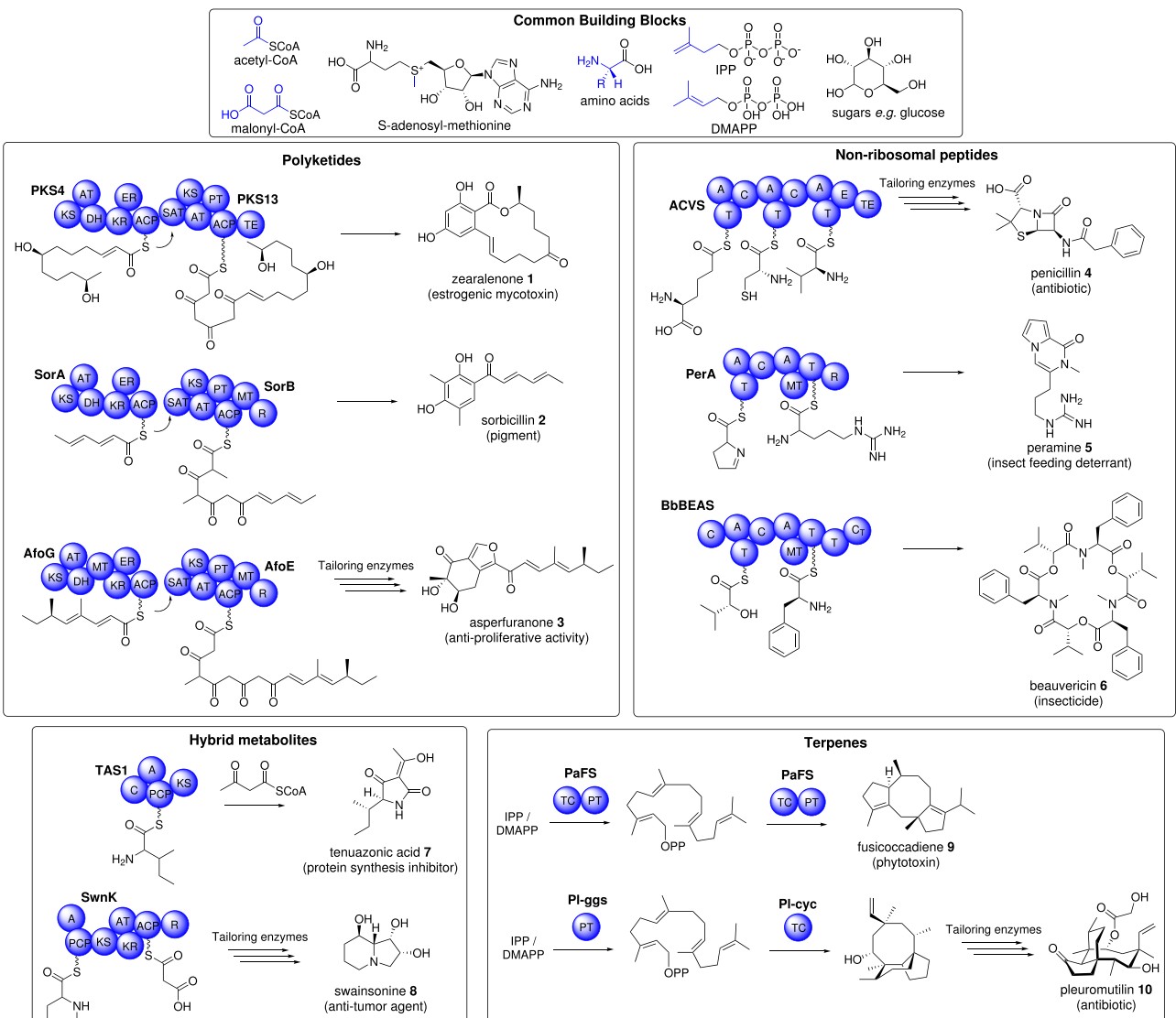

**Fig. 1 | Examples of fungal natural products and the module/domain composition of each megasynth(et)ase that produces them.** Common building blocks supply precursors for the biosynthesis of polyketides, non-ribosomal peptides, hybrid metabolites, and terpenes. Domain abbreviations: KS ketosynthase, AT acyltransferase, DH dehydratase, MT methyltransferase, ER enoylreductase, KR ketoreductase, ACP acyl carrier protein, SAT starter unit acyltransferase, PT product template, TE thiolesterase, R reductase, C condensation, A adenylation, T thiolation (also referred to as PCP peptidyl carrier protein), E epimerase, TC terpene cyclase, PT prenyltransferase.

## Polyketide synthases (PKS)

Polyketides are a large class of natural products constructed from the condensation of acyl-CoA units by polyketide synthase (PKS) enzymes *e.g.* **1–3** (Fig. 1). In fungi, PKS enzymes are typically iterative and so it is near impossible to predict the exact structural features of the encoded carbon backbone. These PKS enzymes can be classified as highly reducing (HR-PKS), partially-reducing (PR-PKS), and non-reducing (NR-PKS) according to the presence or absence of domains that catalyze the reduction of the polyketone functionality[12,13].

**NR-PKS domain swaps.** The starter unit acyl carrier protein transacylase (SAT) domain[14] is unique to NR-PKS enzymes and selects the starter unit *e.g.* acetyl-CoA, malonyl-CoA, and more infrequently propionyl-CoA, hexanoyl-CoA, nicotinyl-CoA, and then transfers the substrate to the ketosynthase (KS) domain, making it an obvious choice for engineering. Many SAT domains studied have demonstrated tolerance for alternative, even unnatural, starter units allowing for an effective strategy in synthesizing unnatural natural products[15]. Swapping of SAT domains between NR-PKS has proved crucial in understanding the specificity of other domains within the NR-PKS, revealing a new function of the SAT domain in dimerization[16] and also leading to the production of novel compounds. For example, swaps of the SAT domain from AfoE, involved in asperfuranone **3** biosynthesis, and the SAT domain from StcA, which is involved in sterigmatocystin biosynthesis, led to the production of a new polyketide **11** (Fig. 2), that utilized a hexanoyl starter unit but remained at the same chain length as **3**[17].

The product template (PT) domain is another domain unique to NR-PKS which is essential for the cyclization and aromatization of the polyketide chain[18,19]. Swap of the PT from ApdA, required for asperthecin biosynthesis, into PKS4, responsible for bikaverin biosynthesis, led to the production of the novel α-pyranoanthraquinone **12**[20]. Further investigations of the PT domain programming identified three residues which, when mutated, converted the cyclization mode - potentially identifying a simpler approach in engineering novel natural products[21].

Combinatorial domain swaps between a series of *C*-methyltransferase (CMeT or MT) domains excised from NR-PKS revealed that the

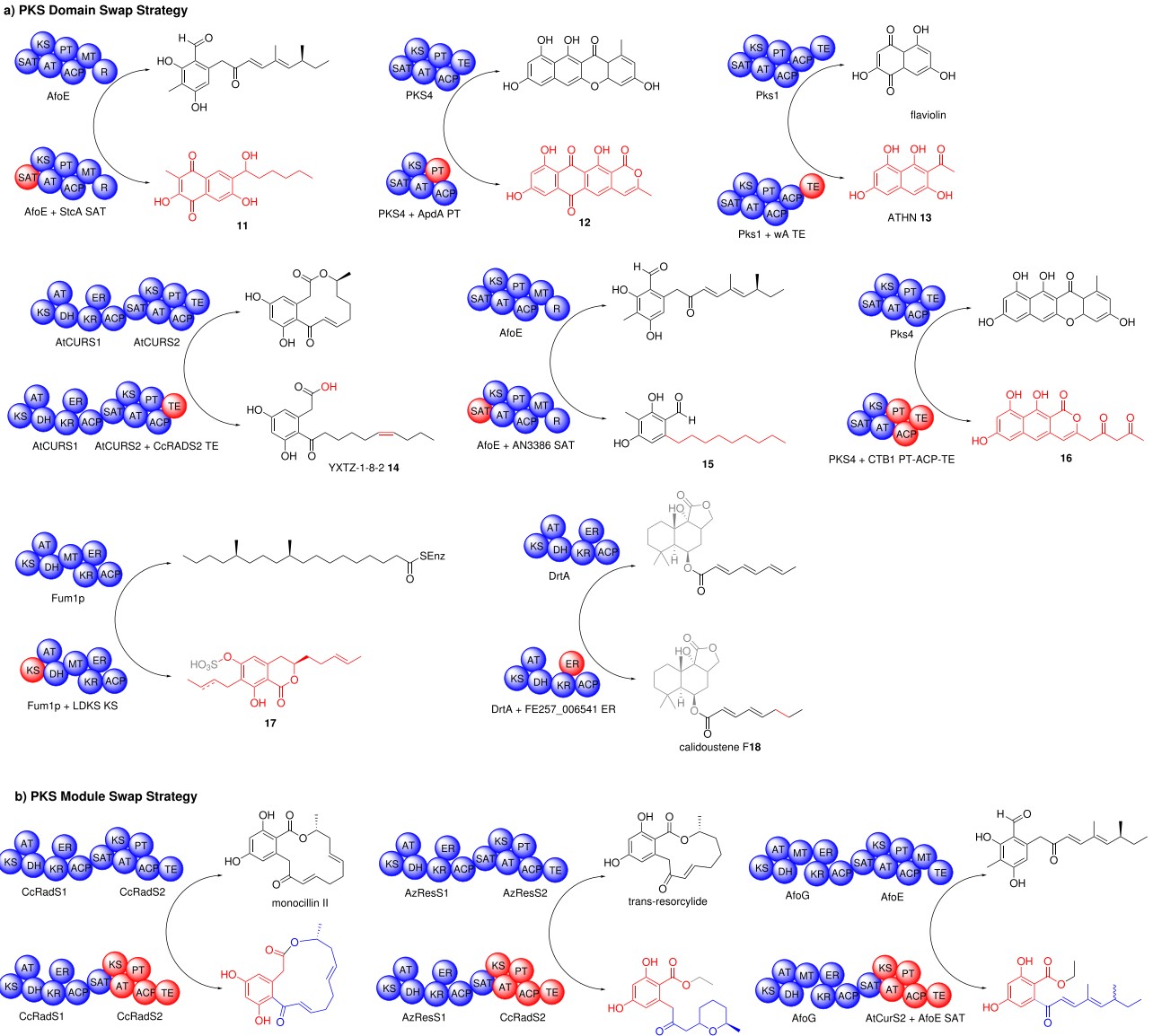

**Fig. 2 | Overview of PKS megasynthase combinatorial strategies in iterative fungal PKS systems and the novel polyketides generated. a** PKS domain swap strategy to construct chimeric enzymes and **b** PKS module swap strategy to construct chimeric enzymes. The native enzyme and its product are shown at the top of the arrow and the engineered enzyme and the new product are shown at the bottom of the arrow. Swapped domains/modules are shown in red. Portions of molecules shown in gray arise from additional enzymes/modifications that are not shown. Domain abbreviations: SAT starter unit acyltransferase, KS ketosynthase, AT acyltransferase, PT product template, MT methyltransferase, R reductase, TE thiolesterase, DH dehydratase, ER enoylreductase, KR ketoreductase, ACP acyl carrier protein.

methylation pattern is inherent to an individual *C*MeT domain, however there is kinetic competition with the KS domain which may override the *C*MeT[22].

Swaps of individual TE domains from wA to Pks1 converted the polyketide product from flaviolin to ATHN **13**, confirming the role of the domain as an unusual bifunctional thiolesterase (TE) that catalyzes both polyketide cyclization and deacetylation[23]. Swapping of the TE domains between the AtCURS1/2 and CcRADS1/2 led to several anticipated and unanticipated macrocycles, pyrones, carboxylic acids, and esters, including YXTZ-1-8-2 **14**[24]. Through swapping both the PT and TE domains concurrently the polyketide products more closely matched those predicted by biosynthetic rules[24]. This study indicated that TE domains act as important decision gates in polyketide release mode and are influenced by the chain length and geometry of the PT-cyclized intermediate being presented for release. Furthermore, swaps of TE domains between relatively close NR-PKS systems can actually lead to increased titers of useful polyketides[25].

Typically, more than one domain is swapped at a time for systematic studies[26]. For example, when the SAT from AfoE was replaced by that from AN3386 a novel major $C_{16}$ polyketide **15** was detected. However, when the SAT-KS-AT, or SAT-KS-AT-PT domains were swapped, the major compound was a $C_{18}$ polyketide, revealing that control of chain length resides within the KS domain[27]. KS domain swaps between NR-PKS CoPKS1, which synthesizes the octaketide atrochrysone carboxylic acid, and CoPKS4, which synthesizes both octaketides and heptaketides such as 6-hydroxymusizin, further corroborated the role of the KS domain in controlling polyketide chain length and identified ten amino acid residues that may be involved[28].

Similarly, individual PT, ACP, and TE domains were swapped with six SAT-KS-AT tridomains resulting in 72 different combinations. The study revealed that the production of new polyketides required cyclization and release to be faster than spontaneous reactions and off-loading, and hypothesized that a matched SAT-KS pair is essential. Seven previously unknown

polyketides were detected including 1-(7,9,10-trihydroxy-1-oxo-1*H*-benzo[*g*]isochromen-3-yl)pentane-2,4-dione **16**[29].

**HR-PKS domain swaps.** Domain swaps in HR-PKS are inherently more challenging due to many lacking a terminal release domain, and the detection of the non-aromatic products being more difficult[30]. The first reported domain swap between HR-PKS systems, was the swap of the KS domain from Fum1p, involved in fumonisin biosynthesis, with PKS1, responsible for T-toxin biosynthesis. The chimeric PKS still produced fumonisins, albeit at a much lower level than the wild-type strain, despite PKS1 encoding a much longer polyketide ($C_{41}$ *vs* $C_{18}$)[31]. In a similar strategy, the KS domain of Fum1p was swapped with the KS domain from LDKS, the diketide synthase responsible for the C4 polyketide side chain of lovastatin. Here, four dihydroisocoumarins were produced, two of which were novel, at considerably higher titers than the previous experiment. Surprisingly all four were aromatic and two contained sulfate groups added *via* the host *e.g.* **17**[32].

The enoylreductase (ER) domain has been swapped in DrtA, the HR-PKS involved in the biosynthesis of fungal drimane-type sesquiterpene esters. These metabolites are meroterpenoids consisting of a sesquiterpene core attached to a highly reduced polyketide of different chain lengths ($C_6$ and $C_8$) and various levels of unsaturation. The donor ER domain came from the HR-PKS believed to be involved in the biosynthesis of fully saturated drimane-type sesquiterpene esters, identified from genome mining and phylogenetic analysis. Over-expression of the chimeric HR-PKS in its native host led to the production of a novel drimane-type sesquiterpene ester, calidoustrene F **18**. However, heterologous expression of the chimeric HR-PKS with other genes from the cluster led to the production of six previously unreported drimane-type sesquiterpene esters with different levels of saturation, including full reduction of the attached octaketide, as determined by HRMS[33].

**PKS Module Swaps.** With the exception of HR-PKS and NR-PKS pairs, which collaborate to synthesize a polyketide product, fungal PKS enzymes are not usually considered as modular enzymes. The first example of combinatorial biosynthesis of fungal PKS modules was the systematic swaps of HR-PKS and NR-PKS pairs that collaborate to synthesize benzenediol lactone (BDL) scaffolds[34]. Four BDL HR-PKS and NR-PKS pairs were methodically swapped to generate twelve unnatural pairs of HR-PKS and NR-PKS enzymes which were heterologously expressed in *Saccharomyces cerevisiae* for polyketide production and analysis. In five of the twelve swaps, the SAT domain of the NR-PKS needed to be exchanged to enable acceptance of the HR-PKS and its polyketide intermediate. Remarkably ten out of the twelve swaps resulted in production of the predicted metabolite and, of these, six were novel natural products *e.g.* **19** and **20**.

In a related study, HR-PKS and NR-PKS pairs were swapped between the asperfuranone PKS and four BDL systems[35]. Although the SAT domain needed to be exchanged in the accepting BDL NR-PKS, three swaps were successful and resulted in the production of four unnatural natural products *e.g.* **21**.

Finally, in a follow-up study, five chimeric NR-PKS enzymes were generated which possessed non-natural SAT or TE domains, or both, and heterologously expressed with non-native HR-PKS partners, generating a total of 35 unnatural enzyme combinations[36]. These unnatural enzyme combinations led to the production of 23 novel compounds. However, the major outcome of the study was understanding the major roles of the SAT and TE domains and how their evolution dictates engineering possibilities.

## Non-ribosomal peptide synthetases (NRPS)
Non-ribosomal peptides are known for their diverse bioactivities such as antifungal, antibacterial and immunosuppresant[37–39], making them valuable in drug development. The non-ribosomal peptide synthetase enzymes that synthesize these peptides consist of condensation (C), adenylation (A), and thiolation (T) - also referred to as a peptidyl carrier protein (PCP) - domains, organized into modules[40]. Some fungal NRPS modules also include

methyltransferase (MT), epimerization (E), or oxidation (OX) domains, and the modules may or may not be iterative.

**NRPS domain swaps.** BbBEAS and BbBSLS are iterative NRPS enzymes with a C-A-T-C-A-MT-T-T-C domain organization, encoding the biosynthesis of beauvericin **6** and bassianolide respectively. Chimeric enzymes were constructed in vitro; where the terminal T-T-C, T-C, or C domains were swapped from BbBSLS into BbBEAS, a new tetrapeptide compound FX1 **22** was observed (Fig. 3)[41]. The role of the tandem T domains was investigated where both were shown to be required for the efficiency of the system but each can function without the other. The roles of the C domains were also investigated and although the first C domain did not possess condensation activity in vitro, it was essential for the overall activity of the NRPS. In contrast, the terminal C domain was shown to have condensation activity as well as control chain length and macrocyclization.

Taking advantage of iterative NRPS systems, chimeric NRPS comprising of domains of hexa- and octa-cyclodepsipeptide (CDP) systems were constructed, resulting in the formation of novel CDPs octa-beauvericin (FX1) *e.g.* **22** and octa-enniatin, on mg scale, showing enhanced anti-parasitic activity compared to clinically used drugs[42]. Furthermore, a chimeric NRPS was able to produce $1.3 \, \text{g} \, \text{L}^{-1}$ of the known peptide hexa-bassianolide, significantly improving reported titers from both synthesis and fermentation.

PpzA-1 is a dimodular NRPS responsible for the biosynthesis of per-amine **5**. PpzA-2 is a truncated dimodular NRPS with similar domain organization to PpzA-1 but lacks an R domain, contains significant mutations, and was originally thought to be non-functional, or the product of a pseudogene. When PpzA-2 was expressed in the fungal host *Penicillium paxilli* a new compound was detected and identified as the pyrrolopyrazine-1,4-dione **23**. Utilizing meiotic recombination events as a (sub)-domain swapping strategy, four different ppzA alleles were generated[43]. In addition to determining that the NRPS could function without the terminal R domain, *via* a non-enzymatic dipeptide release mechanism, this study highlights the power of utilizing allelic diversity to identify and exchange (sub)-domains to generate DKP products.

A strategy has also been developed to swap domains from the fungal iterative cyclodepsipeptide NRPS with the non-iterative, or linear, cyclosporine NRPS. 24 hybrid synthetases were generated and ten successfully led to the production of a peptide natural product enabling exchange rules to be described[44]. Namely, the C domain can be swapped *via* the T-C or C-A linker regions, provided that the C domain is left intact, and the specificity of the swapped C domain is similar, or identical, to substrates accepted by the native system's upstream and downstream modules respectively.

**NRPS module swaps.** The first report of NRPS module swapping utilized the highly similar BbBEAS and BbBSLS systems which synthesize beauvericin **6** and bassianolide respectively. First, the individual modules were expressed in yeast and shown to be functional, albeit at lower titers than the intact enzyme. When modules were swapped between the systems, the product profile was switched according to the specificity of the later modules. The linker between the modules was shown to be essential and the chimeric systems were more efficient when a chimeric linker was used[45].

Two hybrid dimodule NRPS systems were generated by fusing module one from PSYN (PM 1) with the hexadomain module two with ESYN (EM 2) and BYSN (BM 2) respectively. Expression of these chimeric NRPS in *E. coli* led to the production of novel cyclodepsipeptides [PheLac] enniatin **25** and [PheLac]-beauvericin **24**[46].

More complex chimeric NRPS enzymes were created by fusing two and three dimodules to produce tetra- and hexamodule systems. Although the NRPS were functional and led to the production of several novel peptides, unexpectedly the system demonstrated a combination of iterative behavior and module skipping, revealing that the terminal C domain can direct an iterative system to behave as a mixed iterative/linear system[47].

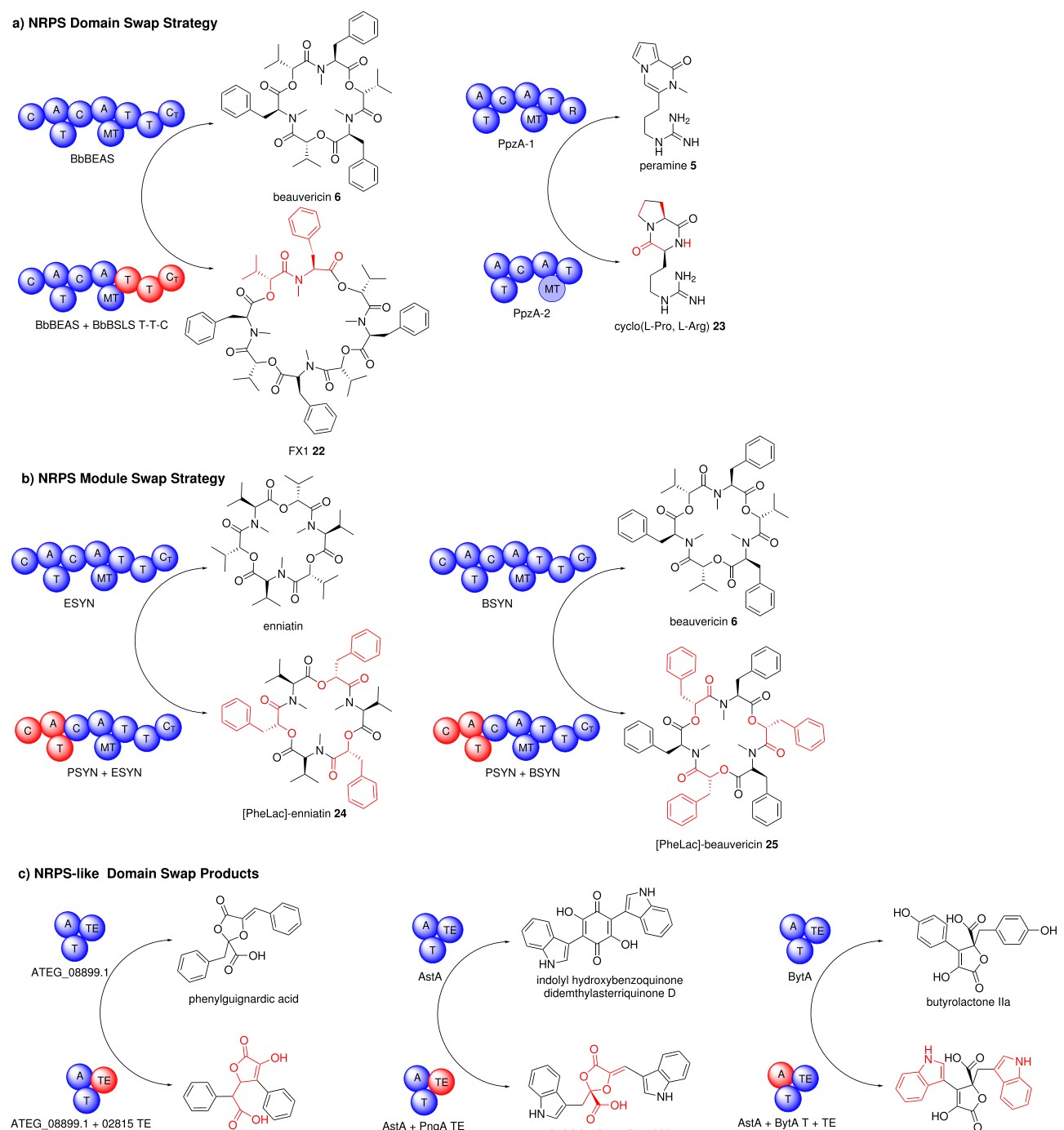

**Fig. 3 | Examples of novel peptides produced by combinatorial biosynthesis *via* engineering NRPS and NRPS-like megasynthetase modules and domains.**
**a** NRPS domain swap strategy to construct chimeric enzymes and **b** NRPS module swap strategy to construct chimeric enzymes. **c** NRPS-like domain swap strategy to construct chimeric enzymes. The native enzyme and its product are shown at the top of the arrow and the engineered enzyme and the new product are shown at the bottom of the arrow. Swapped domains/modules are shown in red. Domain abbreviations: C condensation, A adenylation, T thiolation (also referred to as PCP peptidyl carrier protein), MT methyltransferase.

## Domain swaps in NRPS-like enzymes

NRPS-like enzymes are a subclass of NRPS enzymes which consist of an A, T, and TE or R domain but lack the canonical C domain for peptide bond formation[48]. In the majority of systems studied to date, two identical α-ketoacids are selected and activated by the A domains and dimerized *via* the TE domain. The first report of NRPS-like domain swapping focused on three enzymes from *Aspergillus terreus* which produce aspulvinone E, butyrolactone IIa, and phenguignardic acid respectively[49–54]. The strategy involved swapping the A domains and TE domains independently between

the three systems and heterologously expressing in *Aspergillus nidulans*. Swaps of the A domains were functional and kept the inherent selectivity of the A domain. Similarly, TE domain swaps were also functional and successfully redirected the cyclization of the resulting lactones. Through these swaps the novel phenylbutyrolactone IIa **26** was generated[54].

A related study swapped domains in a more diverse set of NRPS-like enzymes that differed in substrate selection and mode of dimerization. 24 swaps were conducted and 19 were successful, yielding the expected product in titers of 0.01–30.5 mg L$^{-1}$ in their yeast host. Two of the products

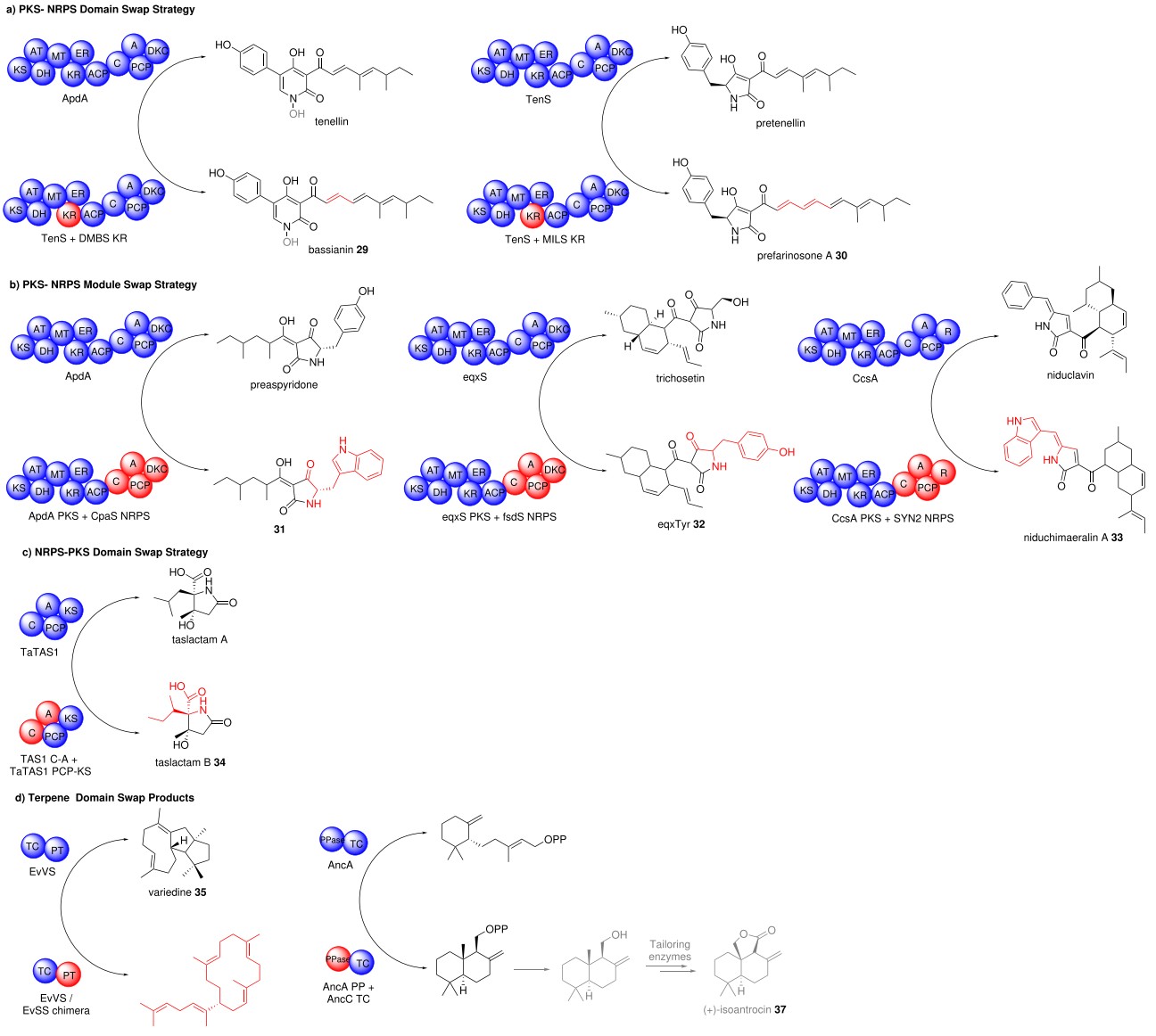

**Fig. 4 | Examples of novel polyketide-derived compounds and terpenes produced by combinatorial biosynthesis *via* engineering megasynthase domains/modules.** **a** PKS-NRPS domain swap strategy to construct chimeric enzymes, **b** PKS-NRPS module swap strategy to construct chimeric enzymes, **c** NRPS-PKS domain swap strategy to construct chimeric enzymes, and **d** terpene domain swap products to construct chimeric enzymes. The native enzyme and its product are shown at the top of the arrow and the engineered enzyme and the new product are shown at the bottom of the arrow. The swapped domains/modules are shown in red. Domain abbreviations: KS ketosynthase, AT acyltransferase, DH dehydratase, MT methyltransferase, KR ketoreductase, ER enoylreductase, ACP acyl carrier protein, C condensation, A adenylation, PCP peptidyl carrier protein, DKC Dieckmann cyclase, TC terpene cyclase, PT prenyltransferase, PPase pyrophosphatase.

generated, indolyl guignardic acid **27** and indolyl butyrolactone **28**, were novel, demonstrating the power of the approach[55]. While some of the swapped TE domains were determined to be more promiscuous than others, precise motifs for substrate specificity or mode of dimerization could not be established from the protein sequence alone.

## PKS-NRPS and NRPS-PKS

In fungi, PKS-NRPS are hybrid enzymes consisting of an iterative HR-PKS fused to a non-iterative NRPS module *via* a flexible linker[56,57]. In contrast to PKS-NRPS enzymes, NRPS-PKS enzymes consist of an NRPS module (A and PCP domains, sometimes with a C domain preceding the A domain) fused to a PKS module (consisting of at least a KS domain, but often times with up to a full HR-PKS)[58]. A subclass of NRPS-PKS includes a C-A-PCP-(PCP)-KS-(TE) domain architecture; the NRPS module selects and activates an amino acid and condenses it with acetoacetyl-CoA, selected by the KS domain.

**PKS-NRPS domain swaps**. The first PKS-NRPS system to be fully interrogated at the domain level was TenS essential for biosynthesis of the pentaketide-derived tenellin. All domains within the HR-PKS module were swapped in different combinations from the comparable domains within DMBS, the PKS-NRPS responsible for the biosynthesis of the closely related hexaketide-derived desmethylbassianin. Through these swaps the KR domain was identified as influencing the chain length of the polyketide, in some experiments leading to production of a hexaketide-derived product as the sole product. Heterologous expression of this chimeric PKS-NRPS with tailoring genes from the tenellin pathway led to the production of bassianin **29**, a known natural product, now extinct as the original producing strain is no longer available (Fig. 4)[59]. In a follow-up study heptaketide-derived products *e.g.* **30** could also be generated by swapping in the KR domain from MILS, the PKS-NRPS required for the biosynthesis of militarinone. The study highlighted that swaps of comparably small sub-fragments of just 12

amino acids within the KR could reprogram the chain length of the whole PKS[60].

**PKS-NRPS module swaps.** Due to the size of these enzymes (~400 kDa) they are extremely challenging to study in vitro. However, the fungal PKS-NRPS, ApdA, was purified as functional protein after expression in *S. cerevisiae*. The PKS module of ApdA and the NRPS module of CpaS were reconstituted in vitro and a novel tetramic acid **31** was formed, albeit in low titers (~0.1 mg L$^{-1}$). Besides being the first study to reconstitute a fungal PKS-NRPS enzyme in vitro, the programming rules controlling substrate specificity, polyketide chain length, methylation, enoylreduction, and protein-protein interactions could be investigated[61].

The rules for successfully swapping NRPS modules were investigated in depth *via* 57 different gene fusion combinations, leading to 34 distinct module swaps. The position of the swap could be at the ACP domain, or the C domain, within the ACP domain, or by introducing a tandem ACP domain. The study defined several crucial programming rules: the PKS module in collaboration with its cognate *trans*-ER synthesizes the polyketide; C domains are highly selective to which substrates they will accept; and ACP domains are interchangeable provided that they are closely related[62]. In total five new polyketide products were generated including eqxTyr **32**, the tyrosine analog of equisetin.

The role of the flexible linker region between the PKS and NRPS modules has also been investigated, indicating that the length of the linker, its sequence, and its removal or replacement with red fluorescent protein (RFP) have little effect on product formation[63]. However, the linker is essential for keeping the PKS and NRPS in close proximity in vivo. The construction of chimeric PKS-NRPS hybrids between CcsA and Syn2 led to the production of two novel compounds including niduchimaeralin A **33**.

Although swapping entire NRPS modules would appear straightforward, it is more challenging to convert an HR-PKS to a PKS-NRPS by fusing an NRPS module. When this was attempted using LovB and the NPRS module for CheA in vivo (within the lovastatin biosynthetic gene cluster) no new products could be detected, and the native product lovastatin was no longer produced[64]. In-depth phylogenetic studies indicated that PKS and NRPS modules in these composite enzymes share an evolutionary history and that domain/module swaps from distantly related enzymes may not be possible. Furthermore, the selection of the amino acid by the NRPS domain may require both the A and C domains.

**NRPS-PKS domain swaps.** Domain swaps between NRPS-PKS systems have so far been conducted in TAS1, which synthesizes tenuazonic acid **7** *via* isoleucine and acetoacetyl-CoA (Fig. 1), and TaTAS1, which synthesizes taslactam A *via* leucine and acetoacetyl-CoA. The swaps demonstrated that the amino acid incorporated by the system could be altered by swapping the C-A didomain *e.g.* taslactam B **34**[65]. Possibly due to differences in the mode of cyclization, swaps of the terminal KS domain did not lead to detectable product formation.

**Terpene synthase (TS) domain swaps**
Terpenoids are a vast class of natural products with numerous pharmacological applications[66]. The biosynthesis of terpenoids typically follows a two-step process: first, terpene cyclases (TCs) create the foundational hydrocarbon structure, and then tailoring enzymes, including cytochrome P450 monooxygenases (P450s), modify the cyclized hydrocarbon by functionalizing C−H bonds. Some terpene synthases in fungi are bifunctional enzymes that consist of a *C*-terminal prenyltransferase (PT) and an *N*-terminal terpene cyclase (TC) that sequentially condense IPP and DMAPP units and then cyclize the backbone. PT and TC domain swaps between two previously cryptic bifunctional terpene synthases, EvVS which produces novel diterpene **35**, and EvSS, enabled the production of a novel sesterterpene **36**[67]. The study demonstrated that the PT domain can synthesize different-length terpene precursors but the TC domain controls which lengths are acceptable.

Another type of bifunctional terpene synthases are found in mushrooms; these enzymes contain pyrophosphatase (PPase) and TC domains. AncC is an example of this type of bifunctional TS which when expressed in yeast with tailoring enzymes leads to the production of (+)-antrocin **37**. Deletion or swapping of the TC domain showed that cyclization activity is abolished or modified, and mutagenesis identified the residues essential for activity the selectivity in both the PPase and TC domains[68].

**Trans-acting enzyme swaps**
An increasing number of fungal megasynth(et)ases have been shown to require the action of collaborating, or *trans*-acting, enzymes[69]. These enzymes are required to ensure fidelity of the programming of the megasynth(et)ases. For example, the role of *trans*-acting enoylreductases (*trans*-ER)[70], ketoreductases (*trans*-KR)[71,72], acyltransferases (*trans*-AT)[73], thiolesterases (*trans*-TE)[74], and oxidases[75], have been investigated by combinatorial biosynthesis. While none of these studies have led to the production of novel compounds yet, they did reveal crucial information regarding the substrate specificity of the *trans*-acting enzymes.

## Combinatorial biosynthesis of tailoring enzymes
Collectively, tailoring enzymes are the enzymes that act upon the carbon backbone synthesized by a megasynth(et)ase enzyme and often introduce significant structural diversity and/or complexity[76]. Examples include various classes of enzymes such as oxidases, reductases, oxidoreductases, transferases, and cyclases. Investigations of biosynthetic tailoring enzymes in unnatural combinations is a relatively rapid method to elucidate enzyme function and often leads to new compounds being produced.

### Novel meroterpenoids and sterols
Decalin-containing diterpenoid pyrones (DDPs) are natural meroterpenoids derived from fungi, characterized by a decalin ring system linked to a pyrone moiety. Recently, five distinct DDP biosynthetic gene clusters (*dpfg*, *dpmp*, *dpch*, *dpma*, and *dpas*) were identified in five different fungi. Despite sharing a common early biosynthetic pathway, leading to a shared intermediate, the divergence in the tailoring enzymes within each pathway results in the production of diverse DDPs. One such enzyme, DpasF from the *dpas* BGC, is a flavin-dependent monooxygenase (FMO) responsible for tetrahydrofuran (THF) ring formation or enone formation at the $C_5$ unit of DPPs. Co-expression of *dpasF* with the *dpmp*/*dpfg* BGC in *A. oryzae* yielded five new DPP analogs, such as **38** and **39** (Fig. 5). Another tailoring enzyme, DpmpI from the *dpmp* BGC, is a methyltransferase that methylates the carboxylic acid of DPPs. Introduction of *dpmpI* into *A. oryzae* with *dpch*/*dpas* BGC led to the production of four new DPP analogs, such as the *O*-methylated higginsianin A **40**[77].

Tropolone sesquiterpenoids are meroterpenoids that share a core 11-membered macrocycle derived from humulene, connected to one or two polyketide-derived tropolones *via* dihydropyran rings[78]. Combinatorial biosynthesis to generate unnatural tropolone sesquiterpenoids was successfully accomplished by combining genes from three different tropolone sesquiterpenoids BGCs (*aspks1* BGC, *eup2* BGC, and *pyc* BGC) in the fungal host *A. oryzae* NSAR1. Through the co-expression of *eup2R6*, a P450 gene, with genes (*aspks1, asL1, asL3, asR2, asR5, asR6*) responsible for xenovulene B production, a novel compound, 10-hydroxyxenovulene B **41**, was generated. Additionally, *eup2R5*, encoding a FAD-dependent monooxygenase, was identified as a catalyst for oxidative ring-contractions when co-expressed with genes producing xenovulene B[79].

Biosynthesis of polytolypin, a fernane-type triterpenoid from fungi, involves the triterpene cyclase PolA, that catalyzes the cyclization of 2,3-oxidosqualene to motiol, and three P450 enzymes that introduce multiple oxidation steps. Isomotiol and fernenol are structurally similar to motiol. When three P450s (*PolB/C/E*) were co-expressed with isomotiol or fernenol cyclase genes in *A. oryzae* NSAR1 respectively, five novel polytolypin analogs resulted such as **42**[80].

Fumagillin is known for its antiangiogenic activity due to binding to human methionine aminopeptidase[81]. In the biosynthesis of fumagillin, the

**Fig. 5 | Examples of novel terpene-derived molecules produced *via* combinatorial biosynthesis *via* swapping tailoring enzymes.** New functional groups and/or structural modifications arising from combinatorial biosynthesis are shown in red.

terpene cyclase Fma-TC catalyzes the conversion of farnesyl-diphosphate (FPP) into β-trans-bergamotene, which is then sequentially oxidized to produce epoxycyclohexanone by the multifunctional P450 enzyme Fma-P450. TC-P450 pairs from *Trichoderma virens (Tv86)* and *Botryotonia cinerea (BC)* are homologous to Fma TC-P450 pairs. Combining Tv86-TC and Fma-P450 pairs, or Fma-TC and Tv86-P450 pairs, in *S. cerevisiae RC01* led to the production of a new group of novel sesquiterpenoids with hydrocarbon scaffolds different from bergamotene including **43** and **44**[82].

PeniB is a multifunctional P450 enzyme that catalyzes crucial tailoring steps in the production of the insecticide penifulvin A, itself derived from a polyquinane sesquiterpenoid (PQST) scaffold. Combining *peniB* with sesquiterpenoid cyclases such as *bcbot2* and *Coll34TC* in *S. cerevisiae* RC01 resulted in a variety of unnatural oxidative products *e.g.* **45**. Similar to PeniB, CYP450-4710 also possesses multifunctional oxidation ability on different PQST scaffolds including the plant-derived PQST scaffold that is synthesized by MrTPS2 from chamomile *e.g.* **46**[83].

Two distinct biosynthetic pathways, designated as *peni* and *aspe*, have been characterized for fungal dioxafenestrane sesquiterpenes (asperaculin A and penifulvin A). AspeD, an α-ketoglutarate-dependent dioxygenase, is a tailoring enzyme in the *aspe* pathway that catalyzes the C7 R-hydroxylation during the biosynthesis of asperaculin A. Interestingly, AspeD exhibits a promiscuous hydroxylation ability towards dioxafenestrane sesquiterpene intermediates derived from the peni pathway, resulting in the production of unnatural products, such as **47**. In contrast, the homologous protein PeniD, originating from the *peni* pathway, demonstrates specificity towards only native substrates[84].

Aristolochene is a bicyclic sesquiterpene synthesized by aristolochene synthase, which is the parent hydrocarbon of a large variety of fungal toxins such as PR toxin and sporogen-AO 1[85,86]. Various oxidized aristolochenes were synthesized through the combination of tailoring enzymes and terpene cyclases sourced from homologous biosynthetic pathways. Co-expression of terpene cyclase gene *hrtc* and p450 genes *prL4* and *prL7* resulted in the production of hydroxylated congeners of PR toxin from *Penicillium roquefortii e.g.* **48**. While co-expression of another p450 gene *xhR1* with *hrTc* provided six new aristolochene congeners *e.g.* **49**[87].

Steroids are widely used clinically as anti-inflammatory, immunosuppressive, and anticancer agents[88]. Hydroxylation of steroids is a pivotal step in the industrial synthesis of diverse steroid drugs. The incorporation of cytochrome P450 CYP103168 and cytochrome P450 reductase CPR64795 from the fungus *Cochliobolus lunatus* into two *Mycolicibacterium smegmatis* mutants, designated MS6039-5941 and MS6039, resulted in the creation of recombinant mutants capable of efficiently converting cholesterol (CHO) and phytosterols (PHYTO) into 14αOH-4-androstene-3,17-dione (14OH-4AD, **50**) and 14αOH-1,4-androstadiene-3,17-dione (14OH-ADD, **51**) with high yields, respectively[59]. Additionally, the introduction of 11α-hydroxylating enzymes from the fungus *Rhizopus oryzae* into the previously mentioned two *M. smegmatis* mutants led to the production of 11αOH-AD **52** and 11αOH-ADD **53** with CHO or PHYTO as feedstock[89].

The co-expression of a CYP enzyme (CYPN2) and a cytochrome P450 reductase (CPRns) from *Nigrospora sphaerica* within *Pichia pastoris* GS115 revealed noteworthy 6β- and 15α-hydroxylation capabilities on progesterone *e.g.* **54**. Moreover, the *Pichia pastoris* GS115 strain carrying CYPN2-CPRns genes demonstrated the ability to convert cortisone, AD, and DHEA, yielding novel products *e.g.* **55, 56**, and **57**. Interestingly, distinct hydroxylation specificities were observed across various steroid substrates[90]. These breakthroughs pave the way for generating hydroxylated steroidal synthons with different specificities at an industrial scale.

## Novel polyketide derivatives

Cytochalasans are natural inhibitors of actin polymerization, and consist of a large family of PKS-NRPS derived metabolites. During investigations into the biosynthesis of cytochalasans, *Magnaporthe grisea* mutants were developed to study the function of cryptic P450s in similar BGCs. Not only could novel cytochalasans be generated *e.g.* **58** (Fig. 6), but the site-specific oxidative functions *i.e.* epoxidation vs. hydroxylation could be determined[91].

Novel statins were generated in *S. cerevisiae* by mixing genes from the lovastatin BGC and FR901512 BGC. Co-expression of the acyltransferase gene (*frlD*) with the *lovABCG* genes yielded a new statin structure, presumably *O*-acetylmonacolin J **59**. Similarly, co-expression of the acyltransferase gene (*lovD*) and polyketide synthase (*lovF*) genes with the

**Fig. 6 | Examples of novel polyketide-derived molecules produced *via* combinatorial biosynthesis via swapping tailoring enzymes.** New functional groups and/or structural modifications arising from combinatorial biosynthesis are shown in red.

*frlABCG* genes produced a new statin structure **60** where the tetralin nucleus was modified with a methylbutyrate side chain[92].

Aurovertins are fungal mycotoxins that have been reported to show strong anticancer activities[93]. The FMO AurC plays an important role in the construction of the dioxabicyclo-octane (DBO) moiety of the aurovertins. A new compound **61**, with a DBO ring moiety that has the same configuration as aurovertin E, was produced in high titers (480 mg L$^{-1}$) when CtvA, the PKS responsible for citreovirdin biosynthesis, was introduced into the *C. arbuscula* Δ*aurA* mutant[94].

Fungal benzenediol lactone (BDL) polyketides, containing a 1,3-benzenediol moiety bridged by a macrocyclic lactone ring, offer rich pharmacophores with broad-ranging bioactivities, such as modulating the heat shock response and the immune system[34]. The substrate tolerance and regiospecificity of two *O*-methyltransferase tailoring enzymes, LtOMT and HsOMT, from fungal BDL biosynthetic pathways were investigated by combinatorial biosynthesis. Co-expressing the HR-PKS/NR-PKS PKS pairs responsible for BDL production with *LtOMT* and/or *HsOMT* resulted in the production of non-native *O*-methylated BDLs, including *ortho-O*-methylated, *para-O*-methylated, and *o,p*-O-dimethylated products *e.g.* **62**–**64**. Remarkably, both *O*-methyltransferases exhibited flexibility in tolerating diverse BDL ring systems, including variations in macrocycle size, functionalization, and even the absence of the macrocyclic ring. LtOMT catalyzed the methylation of the phenolic hydroxyl group positioned at the *ortho* position relative to the aromatic carbon bearing the carbonyl. Conversely, HsOMT demonstrated *O*-methylation at the phenolic hydroxyl group located at the *para* position to the same aromatic carbon. This regioselective *O*-methylation capability of the tailoring enzymes when applied to a broader range of phenolic polyketide natural products, offers distinct advantages over environmentally unfriendly chemical methods that necessitate high pressure and temperature, often resulting in undesired byproducts[95].

Using a similar approach, the substrate tolerance of FgSULT1, a tailoring phenolic sulfotransferase from the plant-pathogenic fungus *Fusarium graminearum PH-1* was explored. By co-expressing the relevant BDL PKS pairs with *FgSULT1* within an *S. cerevisiae* chassis, sulfated derivatives *e.g.* **65** and **66** could be observed. Notably, all the substrates accepted by FgSULT1 shared a common 2,4-dihydroxybenzaldehyde motif, which was identified as necessary but not sufficient for substrate turnover by FgSULT1. These findings establish a foundation for a synthetic biological and enzymatic platform that can be adapted for the production of bioactive, unnatural product sulfates[96].

Through the application of combinatorial biosynthetic strategies, the glycosyltransferase–methyltransferase (GT–MT) pair BbGT86–BbMT85 was shown to exhibit remarkable efficiency in adorning a diverse array of drug-like substrates, encompassing polyketides, anthraquinones, flavonoids, and naphthalenes, with a methylglucose biosynthon. For instance, the co-expression of the HR-PKS / NR-PKS pair LtLasS1–LtLasS2 from *Lasiodiplodia theobromae* with the BbGT86–BbMT85 pair yielded the unprecedented methylglucoside of desmethyl-lasiodiplodin **67**[97]. During the exploration of BbGT86–BbMT85 orthologs in other fungal genomes, four putative GT–MT modules—MrGT-MrMT, IfGT-IfMT, CmGT-CmMT, and CpGT-CpMT—were identified, which exhibited considerable substrate promiscuity towards different classes of natural products. Utilizing kaempferol as a model substrate, the study revealed that the GT–MT modules display distinct regiospecificities for the same substrate. For example, BbGT86–BbMT85 predominantly methylglucosylates kaempferol at 7-OH *e.g.* **68**, whereas IfGT-IfMT and CpGT-CpMT exhibit a clear preference for 3-OH *e.g.* **69**. In contrast, MrGT-MrMT and CmGT-CmMT tend to synthesize a mixture of monoglucoside and/or monomethylglucosides *e.g.* **68–70**[98].

Sorbicillinoids are a large family of fungal secondary metabolites derived from sorbicillin, showing numerous bioactivities such as antimicrobial, antioxidant, and cytotoxic activities[99]. The function and promiscuity of the FMO PcSorD from the sorbicillinoid pathway in *Penicillium chrysogenum*, was confirmed by heterologous expression in *A. oryzae* NSAR1. PcSorD catalyzes a broad scope of reactions including oxidation, epoxidation, intermolecular Diels–Alder-, and Michael-addition- dimerization reactions[100].

Perylenequinones (PQs) have emerged as promising SARS-CoV-2 as entry inhibitors and one of the most potential photodynamic therapy (PDT) agents in medical treatment[101,102]. Among these, representative compounds include cercosporin, phleichrome, and elsinochromes. The biosynthetic pathway of cercosporin has been elucidated, revealing the involvement of Fe(II)/α-KG-dependent oxygenase CTB9, which catalyzes the final step in the formation of the unusual seven-membered methylenedioxy bridge (ring F) crucial for cercosporin's structure. Remarkably, CTB9 exhibits catalytic activity towards different PQ intermediates, leading to the formation of novel products incorporating ring F. A berberine bridge enzyme-like oxidase CTB5 and a laccase-like multicopper oxidase CTB12 are essential for the double coupling of two naphthol intermediates, forming the perylenequinone core. The two paired enzymes share homology with ElcE and ElcG from the BGC of elsinochrome A, respectively. Reprogramming the

CTB BGC by introducing paired genes *elcE* and *elcG* into a *Cercospora sp. JNU001 ΔCTB6/ΔCTB5/ΔCTB12* strain yields unnatural PQs such as cercosporin A **71** and its diastereoisomers that containing ring F. It is worth noting that cercosporin A **71** exhibits significantly reduced dark toxicity and superior photostability while retaining potent photodynamic anticancer and antimicrobial activities[103]. Similarly, replacing *elcE* and *elcG* with *CTB5* and *CTB12* in the biosynthetic gene cluster of elsinochrome A via heterologous expression experiments in *A. nidulans* results in the production of two new PQs[104]. Furthermore, *Ple6*, found in the phleichrome biosynthetic gene cluster, encodes a fasciclin protein that controlling the axial chirality of PQ cores, particularly in the preparation of (P) helical PQs. Expression of *ple6* in a *Cercospora sp. JNU001 ΔCTB9/ΔCTB10* strain leads to the production of a new (P) helical PQ, (-)-calphostin D **72**[103].

## Combination of fungal and non-fungal enzymes to produce natural and unnatural natural products

Plants and bacteria are also major producers of structurally diverse and bioactive molecules. Combinatorial biosynthesis harnessing the potential of both fungal and non-fungal enzymes is an innovative strategy to break down the boundaries between species, allowing for the creation of entirely new chemical entities with diverse structures and distinct bioactivities (Table 1)[105,106]. For example, the biofuel isobutanol (364 mg L$^{-1}$) was reported from *S. cerevisiae* cultivated under aerobic conditions using the bacterial acetolactate synthase and fungal enzymes including a ketol-acid reductoisomerase, a dihydroxyacid dehydratase, an α-ketoacid decarboxylase, and an alcohol dehydrogenase[107].

The novel chlorinated resveratrol **73** (Fig. 7), with strong antimicrobial and antioxidant activity, was biosynthesized by co-expression of the fungal flavin-dependent halogenase (Rdc2) with the plant resveratrol enzymes tyrosine ammonia-lyase (TAL), 4-coumarate:CoA ligase (4CL), and stilbene synthase (STS) in *E.coli*[105]. RadH is a homolog with a similar sequence to Rdc2 but with a different active site architecture[108]. Co-expression of bacterial 4CL, the plant enzyme feruloyl CoA 6′-hydroxylase (F6′H), and the fungal RadH produced the unnatural chlorinated hydroxycoumarin **74** in *E.*

*coli*[108]. Another new monochlorinated anthraquinone **75** was produced by co-expression of *antA-I* pathway with *radH* in *E. coli*[106]. These studies show the importance of the chlorinated derivatized compounds for the improved bioactivities[106,108].

Carminic acid **76** is a *C*-glucosylated octaketide anthraquinone produced by the insect *Dactylopius coccus* and the main constituent of the natural red food dye carmine[109]. The biosynthesis of carminic acid and *C*-glucosylated anthraquinone (dcII), a precursor for carminic acid, requires five-steps catalyzed by a PKS, monooxygenases, and a C-glucosyltransferase. Heterologous production of these compounds in the fungus *Aspergillus nidulans* was explored by co-expressing a type III PKS (AaOKS) from a plant, with the bacterial cyclase and aromatase (ZhuI, ZhuJ) associated with a Type II PKS pathway (Fig. 8). The formed flavokermesic acid anthrone (FKA) is oxidized to flavokermesic acid (FK) and kermesic acid, catalyzed by an unknown *A. nidulans* FMO or P450[110]. A recent study revealed that the oxidation of FKA by *A. nidulans* FMO, AptC, hydroxylation of FK by endogenous hydroxylase Cat5 from *S. cerevisiae*, and 4′-phospho-pantetheinyl transferase (NpgA) from *A. nidulans* are essential for the biosynthesis of carminic acid in *S. cerevisiae*[111]. Conversion to dcII was achieved using the *D. coccus* *C*-glucosyltransferase DcUGT2 (Fig. 8)[110,111]. An analogous study achieved production of dcII by transient expression in *Nicotiana benthamiana* with AaOKS, ZhuI, ZhuJ, and DcUGT2 genes, aided by unknown plant monooxygenases[112].

The cholesterol-lowering drug pravastatin **77** can be produced by stereoselective hydroxylation of the natural product compactin. By introducing the entire compactin BGC from *Penicillium citrinum* into *P. chrysogenum* with a new bacterial cytochrome P450 (CYP105AS1) from *Amycolatopsis orientalis*, the two-step industrial process can be reduced to a single step with this modified biosynthesis[113].

By manipulating NRP biosynthetic pathways in fungi and incorporating non-fungal NRPS-associated enzymes, researchers have created novel peptide derivatives, or elevated concentration of target products[39]. MbtH-like proteins (MLPs) are associated with adenylation domains of NRPS enzymes in bacteria but are not known in fungi. MLP increases the folding,

## Table 1 | Summary of natural and unnatural products produced by inter-kingdom combinatorial biosynthesis studies, grouped according to the heterologous host used

| Heterologous host | Fungal enzyme (and origin) | Non-fungal enzyme (and origin) | Natural/unnatural products | Yield/mg L$^{-1}$ | Biological activities | Ref. |
|---|---|---|---|---|---|---|
| *E. coli* | Rdc2 (*P. chlamydosporia*) | TAL (*S. espanaensis*); 4CL (*A. thaliana*); STS (*A. hypogaea*) | 2-chloro resveratrol **73**[a] | 7 | Antioxidant, anti-fungal, antibacterial | 105 |
| *E. coli* | RadH (*C. chiversii*) | TAL (*S. espanaensis*); 4CL (*S. coelicolor*); F6′H (*I. batatas*) | 8-chloro-7- hydro-xycoumarin **74** | 1.1 | NR | 108 |
| *E. coli* | RadH (*C. chiversii*) | AntA-I (*P. luminescens*) | Neo-chaetomycin **75**[a] | 0.73 | NR | 106 |
| *P. chrysogenum* | MlcA-H, MlcR | CYP105AS1 (*A. orientalis*) | Pravastatin **77** | 6000 | Cholesterol lowering | 113 |
| *A. oryzae* | grifolic acid synthases (*S. bisbyi*) | DCAS (*R. dauricum*) | daurichromenic acid **78** | 1.23 | Anti-HIV, antibacterial | 114,131 |
| *A. oryzae* | grifolic acid synthases (*S. bisbyi*); AscD (*Fusarium sp.*) | DCAS (*R. dauricum*) | 5-chloro daurichromenic acid **79**[a] | 2.06 | antibacterial | 114 |
| *S. cerevisiae* | AptC (*A. nidulans*); Cat5 (*S. cerevisiae*) | AaOKS (A. arborescens); Zhul and ZhuJ (Streptomyces sp); DcUGT2 (*D. Coccus*) | carminic acid **76** | 7.58 | NR | 111 |
| *S. cerevisiae* | CrtI, CrtYB, CrtS (*X. dendrorhous*) | CrtW (*Paracoccus* sp.); crtZ (*P. ananatis*) | Astaxanthin **81** | 29 µg g$^{-1}$ | Antioxidant | 115 |
| *S. cerevisiae* | native enzymes for mevalonate pathway (*S. cerevisiae*); CrtE (*X. dendrorhous*) | MvaS, MvaE (*E. faecalis*); TASY (*T. cuspidate*) | Taxadiene **82** | 215 | NR | 118–120 |

*NR* not reported.

[a]Novel compounds biosynthesized with fungal and non-fungal enzymes.

**Fig. 7 | Compounds synthesized by combining fungal and non-fungal enzymes.** The substituted group or modification in the structure by the non-fungal enzymes are highlighted in red. Compounds **73–77** are polyketides and **78–82** are terpenoids.

**Fig. 8 | Proposed combinatorial biosynthesis pathway of carminic acid in _S. cerevisiae._** KS Type III PKS (AaOKS) from _A. arborescens_, NpgA 4′-phospho-pantetheinyl transferase from _A. nidulans_, CYC cyclase (Zhul) from _Streptomyces_ sp., ARO aromatase (ZhuJ) from _Streptomyces_ sp, GT _C_-glucosyltransferase (DcUGT2) from _D. coccus_, FMO flavin-dependent monooxygenase (AptC) from _A. nidulans_, OX hydroxylase (Cat5) from _S. cerevisiae_.

stability, adenylation activity, and solubility of NRPS enzyme in bacteria. The transformation of MLPs into _P. chrysogenum_ increased the concentration of penicillin and other intermediates in the biosynthetic pathway[39].

Fungal hosts like _Aspergillus_ have been engineered to produce hybrid terpenoids by introducing non-fungal terpene synthases[114]. The heterologous production of daurichromenic acid **78** (DCA) and a novel compound 5-chloro daurichromenic acid **79** in _A. oryzae_ was reported. The intermediate compound of 5-chloro grifolic acid **80** is produced by the combination of fungal grifolic acid synthases (StbA, a PKS; StbC, a UbiA-like prenyltranferase) and a halogenase (AscD). The plant DCA synthase conducts the 6-_endo-trig_ Wacker-type cyclization to yield **79** (Fig. 7). This chlorinated DCA shows enhanced antibacterial activity against Gram-positive bacteria[114] (Table 1).

The red carotenoid astaxanthin is distributed in fish, birds, algae, and yeast; it is used as pigmentation source and shows high antioxidant activity. The genes encoding a desaturase (_crtI_), bifunctional phytoene synthase/lycopene cyclase (_crtYB_), astaxanthin synthase (_crtS_), and a cytochrome P450 reductase (_crtR_) from red yeast _Xanthophyllomyces dendrorhous_ were expressed in _S. cerevisiae_. This led to the production of a low amount of astaxanthin **81**. However, introducing bacterial genes encoding β-carotene ketolase (_crtW_) and hydroxylase (_crtZ_) accumulated much more astaxanthin relative to the transformant co-expressing _crtS_ and _crtR_[115,116] (Table 1).

The complex diterpenoid drug paclitaxel (Taxol) is used for the treatment of ovarian cancer yet needs to be extracted from the bark of Pacific yew (_Taxus brevifolia_). The destruction of three mature trees yields just 1 g

of paclitaxel[117]. Heterologous expression of the complex biosynthetic pathway using _S. cerevisiae_ host could provide a sustainable solution. The biosynthesis of taxadiene **82**, the first metabolite of Taxol is achieved by yeast (_S. cerevisiae_ and _Xanthophyllomyces dendrorhous_), bacteria (_Enterococcus faecalis_) and plant (_Taxus cuspidate_) genes. Geranylgeranyl diphosphate (GGPP) produced by yeast and bacterial genes _via_ the mevalonate pathway, is cyclized by the taxadiene synthase (TASY) from _Taxus cuspidate_[118]. Several studies show the optimization and improved production of **82** and other oxygenated and acylated Taxol precursors in _S. cerevisiae_[118–120], for example _S. cerevisiae_ KM32 strain could produce 215 mg L$^{-1}$ of **82**[120] (Table 1).

## Outlook

The evolutionary origins of fungal biosynthetic pathways indicate natural deviations in gene transcription, DNA repair, gene loss and duplication, and horizontal gene transfer leading to a constant state of chemical flux and expanded chemodiversity[121]. Similarly, many fungal biosynthetic pathways produce more than one natural product, often referred to as intermediates or shunt metabolites, suggesting that combinatorial biosynthesis naturally occurs within the cell[122]. This is further exemplified by branching and converging biosynthetic pathways frequently found in fungi[123], implying that currently elucidated pathways will continue to evolve and produce variants of known natural products.

As the cost of genome sequencing and synthetic DNA continue to decrease, more chassis organisms and synthetic biology tools are developed for investigating fungal BGCs[124,125], and advances in artificial intelligence

and machine learning continue to emerge[126–128], certainly a greater number of fungal genomes will be mined[129], and the encoded biosynthetic pathways can be investigated more rapidly. This ultimately will lead to a better understanding of the full repertoire of natural products produced by an individual fungus and enable accelerated combinatorial biosynthesis leading to libraries of molecules with desired structural modifications.

Although combinatorial biosynthesis provides a viable alternative to complex multi-step chemical synthesis and the inevitable scale-up required, there are still many challenges ahead, similar to those observed for bacterial biosynthetic enzymes[10,130]. Currently, painstaking studies contrasting and modifying similar systems have been the driving force for revealing the chemical logic within these systems. In some instances, these studies have highlighted that only sub-domain regions are required for re-directing the product profile, perhaps indicating the precision engineering of domains may be more effective than drastic domain replacements[21,42,44,60,72]. Regardless of the approach taken, fundamental understanding of how active sites are occupied in different megasynth(et)ases and how competing catalytic cycles are regulated will be required to successfully redirect intermediates to artificial domains for unnatural processing. Similarly, a deeper understanding of how mature products are retained or released from native enzymes is essential for successfully controlling product release in chimeric systems. Furthermore, how different enzyme types co-ordinate at the domain and megasynth(et)ase level, and with tailoring enzymes, must be explored in much greater detail. While structural biology tools and biochemical assays may help answer some of these outstanding questions at the domain level, techniques to work with complete intact megasynth(et)ases are still lacking due to the challenging size of these systems. Likewise, understanding localization and inter-cellular trafficking of multiple co-ordinating biosynthetic enzymes in native hosts will probably reveal additional considerations for ensuring efficiency in non-native cellular environments.

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

## Acknowledgements
This work was financially supported by UNT Department of Chemistry and BioDiscovery Institute start-up funds, the National Science Foundation (NSF; award number 2048347) and the W. M. Keck Foundation.

## Author contributions
E.S. defined the scope of the review, wrote sections, prepared figures, edited, proofread. S.R. and L.L. also contributed to writing, preparing figures/tables, editing, and proof-reading.

## Competing interests
The authors declare no competing interests.
