## [Peer Review File · Communications Chemistry]

REVIEWERS' COMMENTS:

Reviewer #1 (Remarks to the Author):

This review has comprehensively summarized the advances of combinatorial biosynthesis in the field of fungus-derived natural products. Though this strategy has been widely accepted and applied in the bacterial community, much less work and more difficulties are in the fungal field, somehow due to genetics and different biosynthetic logic. This work overviews different methods on different types of fungal natural products, such as domain/module swapping mainly on PK, NRP and terpenoids, to highly expand the diversity of fungal natural products and create some new unnatural compounds, promisingly for downstream development. This review has been well written and in a very clear logic.

Here are some minor revision suggested:

1. Though the strategies and resulting compounds from combinatorial biosynthesis have been clearly described, should some possible rules be summarized (such as which kinds of domains can be swapped more successfully or more fruitful)? it would be much more beneficial for future development of unnatural natural products with enzyme design and synthetic biology tools, and in a more precise prediction of compound structures and engineering success.
2. Line 31, hybrid molecules are in more diversity, not only NRP-PK, but also PK-terpene, NRP-terpene, and some others with sugar moieties.
3. Line 33, other units, such as sugars can also be the building blocks for fungal natural products.
4. All references should be reformatted uniformly, eg. abbreviations, italic.

Reviewer #2 (Remarks to the Author):

This is a very comprehensive review that provided an insightful overview of the current state of the art on combinatorial biosynthesis of fungal natural product pathways. The breath of the literature it covered is great and will be useful for readers/students who are new to the field as well as experienced researchers who would like to keep on top of the rapidly growing body of work in this area. I recommend this paper for publication with some minor suggestions below:

1. Sometimes it could be difficult to follow some reactions, because the final molecule generated by combinatorial biosynthesis was shown, but unable to find the molecules from the original pathways. However, understandably, the review as it is, already contains a lot of structures and figures.
2. on Page 13 line 378, could cite and discuss this work, in which combinatorial expression of two homologous biosynthetic pathways generated a batch of sesquiterpenoids - Wei et al. ACS Catal. 2021, doi.org/10.1021/acscatal.0c05319".
3. The authors can consider also include the work on the combinatorial biosynthesis of elsinochrome. Hu et al. Chem Sci. 2019, doi.org/10.1039/C8SC02870B in which combinatorial pathway with elc and CTB genes produced novel compounds, and more recently in Su et al. 2024 Angew Chem Int Ed. "Synthetic Biology-based Construction of Unnatural Perylenequinones with Improved Photodynamic Anticancer

Reviewer #3 (Remarks to the Author):

This useful review by Prof. Skellam and colleagues summarizes various efforts to produce unnatural natural products using fungal biosynthetic genes/enzymes in a combinatorial biosynthetic framework. The authors describe megasynthase (PKS, NRPS, composite enzymes, and terpene synthases) engineering by domain and module swaps; detail combinatorial tailoring by noncognate enzymes introduced to biosynthetic pathways; and review biosynthetic pathways assembled from fungal, bacterial, plant and (occasionally) animal-derived enzymes.

The manuscript is mostly lucid and well (although sometimes a bit sparsely) illustrated. The engineering examples are well selected, and provide a comprehensive view of what has up till now been accomplished in this field. This reviewer supports publication after some, mostly minor, clarifications and adjustments.

Line 17: nutraceutical (not neutraceutical)

Line 20, Line 58, Line 62 etc.: The authors often refer to combinatorial synthesis of enzymes and pathways. While technically correct, the point is presumably the combinatorial biosynthesis of natural product analogues/derivatives, not the engineering of hybrid enzymes or enzyme assemblies per se. Line 20: combinatorial biosynthesis of novel products using fungal enzymes...; Line 58: combinatorial biosynthesis with fungal enzymes; etc.

Line 32: delete the word "usually"

Figure 1: In the Polyketides panel, the intermediate on PKS13 is shown after PT-catalyzed first ring cyclization, while in the other two examples (SorB and AfoE) the intermediate is still the linear polyketide chain. This may be confusing as it can suggest the intact incorporation of a dihydroxybenzoic acid precursor by PKS13 for those readers not immersed in PKS literature. In the Terpenes panel, the TC and PT domains are rendered as filled circles and not 3D orbs as for all other domains: this representation may at first suggest that these domains are nonfunctional or not relevant...

Line 40: The figure shows domain/module compositions for individual enzymes, not a generic representation of the "class" that the Legend implies. Simply delete "class".

Line 42: acyl transferase or acyltransferase (as in Line 41) - please be consistent

Line 48: certain compound being produced only in low amounts by strains

Line 79: The abbreviation was defined previously as NR-PKS, not NRPKS.

Line 100, Line 204, Line 273 etc.: "as above" - Maybe as in Fig. 1 Legend?

Line 120: Please indicate the products of CoPKS1 and CoPKS4

Line 141, Line 259: HR-PKS

Line 181: ... but each can function ...

Lines 195-200: This paragraph should be combined and harmonized with the first paragraph (Lines 177-185).

Line 220-221: "selection module" and "extension module" are not good descriptors. Both modules select a substrate and take part in the iterative extension process.

Line 221: pentadomain - correct only if CT is not included in module 2

Line 226: the system demonstrated a combination of iterative behavior and module skipping

Line 230: Domain swaps in NRPS-like enzymes

Line 259, Line 261: Describing tenellin and desmethylbassianin as a pentaketide and a hexaketide is misleading somewhat. They do contain such motifs of course.

Line 265, Fig. 4: Compound number 30 is missing from Fig. 4.

Fig. 4: Why do the authors switch to the use of PCP instead of T (as in Fig. 3)? For readers less familiar with the subject, it is hard enough to keep all the abbreviations in mind...

Line 284: via 57 different gene fusions

Line 286: within, not within in

Line 297: PKS-NRPS (not NRKS)

Line 299: the native product lovastatin (instead of "previously detected")

Line 300: PKS and NRPS modules in these composite enzymes

Line 307: Per Fig. 4, the amino acid constituent in taslactam A and C seems to be the same with only the cyclization different between the two analogues. Thus, this does not seem to illustrate the premise of "amino acid incorporated ... could be altered"

Line 319: As per the figure, compound 35 is the product of a native enzyme, and not that of a swap as the text asserts.

Line 325: The text indicates that "cyclization activity is abolished or modified". Yet, the figure shows the same cyclized intermediate being produced by the native and the hybrid enzymes.

Line 397: cytochrome P450 reductase CPR64795

Line 413: polyketide derivatives

Line 536-537: chloro-grifolic acid is not shown on the figure, so this sentence is not easy to follow, and its connection to DCA biosynthesis is not clear. 5-chloro-grifolic acid is produced, not "achieved".

Lines 538-539: The DCA synthase cannot correspond to a cyclization reaction, but conducts it...

Lines 579-581: This sentence is rather unclear and needs to be edited for syntax. It is not clear what "rates" are referred to either.

Line 587: due to the challenging size

Line 588: Does "mobility" refer to the intracellular trafficking of the enzyme, or to the movement of the domains (and intra-domain structural elements) during catalysis?

Line 589: What is meant by "unnatural cellular environments"?

We are hugely indebted to the three reviewers who took the time to evaluate our manuscript and gave us useful suggestions on how to improve our review. Please see below for our point-by-point responses (in red) to the reviewers comments and suggestions.

Reviewer #1 (Remarks to the Author):

This review has comprehensively summarized the advances of combinatorial biosynthesis in the field of fungus-derived natural products. Though this strategy has been widely accepted and applied in the bacterial community, much less work and more difficulties are in the fungal field, somehow due to genetics and different biosynthetic logic. This work overviews different methods on different types of fungal natural products, such as domain/module swapping mainly on PK, NRP and terpenoids, to highly expand the diversity of fungal natural products and create some new unnatural compounds, promisingly for downstream development. This review has been well written and in a very clear logic.

Thank you very much for your positive remarks and enthusiasm. We appreciate your suggestions and have incorporated them as described below.

Here are some minor revision suggested:

1. Though the strategies and resulting compounds from combinatorial biosynthesis have been clearly described, should some possible rules be summarized (such as which kinds of domains can be swapped more successfully or more fruitful)? it would be much more beneficial for future development of unnatural natural products with enzyme design and synthetic biology tools, and in a more precise prediction of compound structures and engineering success. While we are in agreement that rules would certainly be beneficial to the engineering community, this field is still in its infancy and there are only a few examples of successes with the diverse fungal enzymes. Therefore, we feel it would be premature to define what the rules are when they are still being established.
2. Line 31, hybrid molecules are in more diversity, not only NRP-PK, but also PK-terpene, NRP-terpene, and some others with sugar moieties. We have updated the different classes of natural products and hybrids to include those with sugar moieties.
3. Line 33, other units, such as sugars can also be the building blocks for fungal natural products. We have updated the different classes of natural products and hybrids to include those with sugar moieties.
4. All references should be reformatted uniformly, eg. abbreviations, italic. All the references are checked and corrected.

Reviewer #2 (Remarks to the Author):

This is a very comprehensive review that provided an insightful overview of the current state of the art on combinatorial biosynthesis of fungal natural product pathways. The breath of the literature it covered is great and will be useful for readers/students who are new to the field as well as experienced researchers who would like to keep on top of the rapidly growing body of work in this area. I recommend this paper for publication with some minor suggestions below:

Thank you for your enthusiastic summary and positive remarks. We hope that the readers find it a useful starting point for advancing the field. We appreciate your suggestions and have incorporated them as described below.

1. Sometimes it could be difficult to follow some reactions, because the final molecule generated by combinatorial biosynthesis was shown, but unable to find the molecules from the original pathways. However, understandably, the review as it is, already contains a lot of structures and figures. As per

journal guidelines, we were somewhat limited in page and figure allowances. We tried our best to summarize key highlights and hope that our revisions have made the overall manuscript easier to follow.

2. on Page 13 line 378, could cite and discuss this work, in which combinatorial expression of two homologous biosynthetic pathways generated a batch of sesquiterpenoids - Wei et al. ACS Catal. 2021, doi.org/10.1021/acscatal.0c05319". We have added the related content from this work to Novel Meroterpenoids and Sterols section.

3. The authors can consider also include the work on the combinatorial biosynthesis of elsinochrome. Hu et al. Chem Sci. 2019, doi.org/10.1039/C8SC02870B in which combinatorial pathway with elc and CTB genes produced novel compounds, and more recently in Su et al. 2024 Angew Chem Int Ed. "Synthetic Biology-based Construction of Unnatural Perylenequinones with Improved Photodynamic Anticancer Activities" doi.org/10.1002/anie.202317726. We have added the related work from the two papers to Novel polyketide derivatives

Reviewer #3 (Remarks to the Author):

This useful review by Prof. Skellam and colleagues summarizes various efforts to produce unnatural natural products using fungal biosynthetic genes/enzymes in a combinatorial biosynthetic framework. The authors describe megasynthase (PKS, NRPS, composite enzymes, and terpene synthases) engineering by domain and module swaps; detail combinatorial tailoring by noncognate enzymes introduced to biosynthetic pathways; and review biosynthetic pathways assembled from fungal, bacterial, plant and (occasionally) animal-derived enzymes. The manuscript is mostly lucid and well (although sometimes a bit sparsely) illustrated. The engineering examples are well selected, and provide a comprehensive view of what has up till now been accomplished in this field. This reviewer supports publication after some, mostly minor, clarifications and adjustments.

Thank you very much for your positive remarks and suggestions for improvement. We have incorporated the suggested clarifications and adjustments as described below.

Line 17: nutraceutical (not neutraceutical)- corrected as nutraceutical.

Line 20, Line 58, Line 62 etc.: The authors often refer to combinatorial synthesis of enzymes and pathways. While technically correct, the point is presumably the combinatorial biosynthesis of natural product analogues/derivatives, not the engineering of hybrid enzymes or enzyme assemblies per se. Line 20: combinatorial biosynthesis of novel products using fungal enzymes...; Line 58: combinatorial biosynthesis with fungal enzymes; etc. updated to address well-made points

Line 32: delete the word "usually" deleted

Figure 1: In the Polyketides panel, the intermediate on PKS13 is shown after PT-catalyzed first ring cyclization, while in the other two examples (SorB and AfoE) the intermediate is still the linear polyketide chain. This may be confusing as it can suggest the intact incorporation of a dihydroxybenzoic acid precursor by PKS13 for those readers not immersed in PKS literature. In the Terpenes panel, the TC and PT domains are rendered as filled circles and not 3D orbs as for all other domains: this representation may at first suggest that these domains are nonfunctional or not relevant... figure updated as suggested.

Line 40: The figure shows domain/module compositions for individual enzymes, not a generic representation of the "class" that the Legend implies. Simply delete "class". deleted

Line 42: acyl transferase or acyltransferase (as in Line 41) - please be consistent we have updated to use acyltransferase throughout

Line 48: certain compound being produced only in low amounts by strains added "only"

Line 79: The abbreviation was defined previously as NR-PKS, not NRPKS. Checked all NR-PKS are abbreviated correctly, also confirmed that HR-PKS are abbreviated consistently

Line 100, Line 204, Line 273 etc.: "as above" - Maybe as in Fig. 1 Legend? All legends have been updated to include full abbreviations as per journal guidelines.

Line 120: Please indicate the products of CoPKS1 and CoPKS4 updated

Line 141, Line 259: HR-PKS fixed

Line 181: ... but each can function ... included "each" in the sentence

Lines 195-200: This paragraph should be combined and harmonized with the first paragraph (Lines 177-185). The paragraph is moved to follow on from the first paragraph as suggested.

Line 220-221: "selection module" and "extension module" are not good descriptors. Both modules select a substrate and take part in the iterative extension process. Excellent point, the description has been updated as follows: " Two hybrid dimodule NRPS systems were generated by fusing module one from PSYN (PM 1) with the hexadomain module two with ESYN (EM 2) and BYSN (BM 2) respectively. Expression of these chimeric NRPS in *E. coli* led to the production of novel cyclodepsipeptides [PheLac] enniatin 25 and [PheLac]-beauvericin 24⁴⁶. "

Line 221: pentadomain - correct only if CT is not included in module 2 changed to hexadomain

Line 226: the system demonstrated a combination of iterative behavior and module skipping sentence structure corrected

Line 230: Domain swaps in NRPS-like enzymes section title updated

Line 259, Line 261: Describing tenellin and desmethylbassianin as a pentaketide and a hexaketide is misleading somewhat. They do contain such motifs of course. changed language to be more specific that they are polyketide-derived compounds as follows: "The first PKS-NRPS system to be fully interrogated at the domain level was TenS essential for biosynthesis of the pentaketide-derived tenellin. All domains within the HR-PKS module were swapped in different combinations from the comparable domains within DMBS, the PKS-NRPS responsible for the biosynthesis of the closely related hexaketide-derived desmethylbassianin. Through these swaps the KR domain was identified as influencing the chain length of the polyketide, in some experiments leading to production of a hexaketide-derived product as the sole product."

Line 265, Fig. 4: Compound number 30 is missing from Fig. 4. The compound number has been added to figure 4.

Fig. 4: Why do the authors switch to the use of PCP instead of T (as in Fig. 3)? For readers less familiar with the subject, it is hard enough to keep all the abbreviations in mind... We do agree with this sentiment, however the fields appeared to evolve independently and adopted different usage. We wanted to use the terms described in the original literature in case a reader wants to follow up on a particular study.

Line 284: via 57 different gene fusions the sentence structure has been updated

Line 286: within, not within in corrected

Line 297: PKS-NRPS (not NRKS) fixed typo

Line 299: the native product lovastatin (instead of "previously detected") corrected sentence structure

Line 300: PKS and NRPS modules in these composite enzymes updated

Line 307: Per Fig. 4, the amino acid constituent in taslactam A and C seems to be the same with only the cyclization different between the two analogues. Thus, this does not seem to illustrate the premise of "amino acid incorporated ... could be altered" We have updated the text to be more specific and the figure to show the product with an altered amino acid, as follows: " Domain swaps between NRPS-PKS systems have so far been conducted in TAS1, which synthesizes tenuazonic acid **7** via isoleucine and acetoacetyl-CoA (Figure 1), and TaTAS1, which synthesizes taslactam A via leucine and acetoacetyl-CoA. The swaps demonstrated that the amino acid incorporated by the system could be altered by swapping the C-A didomain e.g. taslactam B **34**⁶⁵."

Line 319: As per the figure, compound 35 is the product of a native enzyme, and not that of a swap as the text asserts. compound 35 is shown as the product of the native EvVS in Fig 4; updated text to make clearer as follows: "PT and TC domain swaps between two previously cryptic bifunctional terpene synthases, EvVS which produces novel diterpene **35**, and EvSS, enabled the production of a novel sesterterpene **36**⁶⁷."

Line 325: The text indicates that "cyclization activity is abolished or modified". Yet, the figure shows the same cyclized intermediate being produced by the native and the hybrid enzymes. We have updated Figure 4 to make the results of the study clearer.

Line 397: cytochrome P450 reductase CPR64795 corrected as suggested

Line 413: polyketide derivatives corrected as suggested

Line 536-537: chloro-grifolic acid is not shown on the figure, so this sentence is not easy to follow, and its connection to DCA biosynthesis is not clear. 5-chloro-grifolic acid is produced, not "achieved". The structure of 5-chloro grifolic acid is shown in the fig 7. The sentences are corrected as, "The heterologous production of daurichromenic acid **78** (DCA) and a novel compound 5-chloro daurichromenic acid **79** in *A. oryzae* was reported. The intermediate compound of 5-chloro grifolic acid **80** is produced by the combination of fungal grifolic acid synthases (StbA, a PKS; StbC, a UbiA-like prenyltransferase) and a halogenase (AscD). The plant DCA synthase conducts the 6-endo-trig Wacker-type cyclization to yield **79** (Figure 7)"

Lines 538-539: The DCA synthase cannot correspond to a cyclization reaction, but conducts it... corrected as "The plant DCA synthase conducts the 6-endo-trig Wacker-type cyclization to yield **79**"

Lines 579-581: This sentence is rather unclear and needs to be edited for syntax. It is not clear what "rates" are referred to either. Clarified as follows: "Regardless of the approach taken, fundamental understanding of how active sites are occupied in different megasynth(et)ases and how competing catalytic cycles are regulated will be required to successfully redirect intermediates to artificial domains for unnatural processing.

Line 587: due to the challenging size corrected

Line 588: Does "mobility" refer to the intracellular trafficking of the enzyme, or to the movement of the domains (and intra-domain structural elements) during catalysis? Line 589: What is meant by "unnatural cellular environments"? Clarified as follows: " Likewise, understanding localization and inter-cellular trafficking of multiple co-ordinating biosynthetic enzymes in native hosts will probably reveal additional considerations for ensuring efficiency in non-native cellular environments."